# EQUIVARIANT GRAPH NEURAL ODEs FOR MODELING PHYSICAL DYNAMICS

## ABSTRACT

Modeling 3D dynamical systems is a fundamental challenge in the physical and engineering sciences, where Equivariant Graph Neural Networks (EGNNs) have emerged as a powerful paradigm by incorporating geometric symmetries. However, these models are fundamentally constrained by their discrete-time, Markovian framework, which neglects long-range temporal correlations and inevitably leads to error accumulation in long-horizon forecasting. To address this limitation, we introduce the *Equivariant Graph Neural Ordinary Differential Equation* (`EG-NODE`), a novel framework that directly learns the continuous-time evolution laws of physical systems. Instead of predicting discrete future states, `EG-NODE` leverages an equivariant GNN as its core to directly model the ordinary differential equation governing the system's instantaneous rate of change the physical *laws of motion* thereby natively preserving SE(3) symmetry within the learning process. This continuous-time paradigm enables high-precision predictions at arbitrary time points and allows for the use of adaptive step-size solvers to dynamically balance computational efficiency and accuracy. Extensive experiments on N-body, molecular, and fluid dynamics benchmarks demonstrate that `EG-NODE` significantly outperforms existing discrete models in long-horizon prediction accuracy and effectively suppresses error propagation. Our work establishes a more fundamental, first-principles-based paradigm for learning continuous physical laws from data. Our codes are available at https://anonymous.4open.science/r/EG-NODE

## 1 INTRODUCTION

The accurate simulation of 3D dynamical systems is a core driver of scientific discovery and engineering innovation (Ghadami & Epureanu, 2022; Brunton & Kutz, 2022; Wang et al., 2023). From molecular dynamics that reveal the binding processes of drugs to target proteins (Decherchi & Cavalli, 2020; De Vivo et al., 2016), to interatomic interaction simulations for designing new materials (Olson, 2000), and fluid dynamics (Wu et al., 2024b;a; Thapa et al., 2020) for climate forecasting (Wu et al., 2025; Gao et al., 2025; Bi et al., 2023), a deep understanding of physical system evolution is crucial. While traditional first-principles numerical simulations are precise (Segall et al., 2002), their prohibitively high computational cost severely limits the scale and speed of exploration. Consequently, developing AI surrogate models that offer both speed and accuracy has emerged as a highly promising research frontier.

Among AI surrogate models, Graph Neural Networks (GNNs) (Corso et al., 2024; Wu et al., 2020) stand out for their natural ability to represent the relational structures between particles. A more critical physical inductive bias is *equivariance* (Lenc & Vedaldi, 2015)the model's symmetry with respect to rigid transformations such as rotations and translations. Enforcing SE(3) equivariance ensures that the physical laws learned by the model are consistent across different coordinate frames, leading to exceptional data efficiency and generalization. (Wu et al., 2023; Xu et al., 2024b) Centered on this principle, works led by E(n)-Equivariant Graph Neural Networks (EGNNs) (Satorras et al., 2021)have significantly advanced the field by embedding this geometric symmetry into their architecture, achieving state-of-the-art performance on multiple benchmarks.

However, despite their success, current mainstream equivariant GNNs are constrained by a shared, fundamental limitation: their **discrete-time, Markovian framework**. In essence, these models learn a mapping from the current state to a future state after a fixed time step. This is akin to learning a

*numerical integrator* rather than the *underlying continuous laws* of the system's evolution. To make long-horizon predictions, the model must be iterated autoregressively, a process known as rollouts. Minute errors introduced at each step **exponentially accumulate** over time, eventually causing the predicted trajectory to catastrophically diverge from the true physical process. Furthermore, the model is "bound" by the discrete timesteps inherent in the training data, lacking the flexibility to make predictions at arbitrary points in time. This fundamental shortcoming severely hinders the application of these models to real-world scientific and engineering problems that demand high-precision, long-horizon forecasting.

To fundamentally overcome this bottleneck, we introduce a novel paradigm: the *Equivariant Graph Neural Ordinary Differential Equation* (`EG-NODE`). Our core idea is to shift from learning discrete integration steps to learning the **ordinary differential equation itself**, which governs the system's continuous evolution. Specifically, we model the system's state evolution as an equation describing its instantaneous rate of change. The key insight is that this *dynamics function*, which dictates how the system continuously evolves, can be parameterized by a **core equivariant GNN**. In this way, `EG-NODE` seamlessly fuses SE(3) symmetry with continuous-time dynamics. We can leverage any advanced ODE solver to integrate these learned "laws of motion," obtaining the system's state over arbitrary time scales, thereby effectively suppressing error accumulation and achieving temporal flexibility. Our main contributions are summarized as follows:

❶ We are the first to propose `EG-NODE`, a novel framework that combines equivariant GNNs with Neural ODEs to learn the continuous-time evolution laws of physical systems.

❷ We demonstrate how to parameterize the dynamics function with an equivariant GNN, ensuring the learned ODE inherently satisfies SE(3) symmetry and is thus more aligned with first principles.

❸ Through extensive experiments on a series of challenging benchmarks, including N-body, molecular, and fluid dynamics, we show that `EG-NODE` significantly outperforms existing state-of-the-art discrete models in long-horizon prediction accuracy.

## 2 RELATED WORK

Our work, the Equivariant Graph Neural Ordinary Differential Equation (`EG-NODE`), is situated at the intersection of three prominent research areas: Graph Neural Networks for physical simulation, E(n)-equivariant deep learning, and Neural Ordinary Differential Equations. This section reviews the core ideas and limitations of these fields to clarify the unique contributions of our work.

**Graph Neural Networks for Physical Simulation.** Graph Neural Networks (GNNs) provide a natural framework for representing multi-particle interaction systems (Fan et al., 2019; Corso et al., 2024; Sanchez-Gonzalez et al., 2018; Pfaff et al., 2020), where particles are nodes and interactions are edges. Pioneering works, such as Interaction Networks, demonstrate the significant potential of GNNs in learning complex physical dynamics via message passing (Brandstetter et al., 2022; Gilmer et al., 2017). However, while these models are permutation equivariant, they generally lack geometric equivariance to Euclidean transformations like rotations and translations (Xu et al., 2024a; Köhler et al., 2019; Wu et al., 2023). They often rely on invariant features such as relative distances, which limits their ability to capture the complete geometric information of a system and sacrifices valuable data efficiency (Kostic et al., 2024; Rame et al., 2022; Li et al., 2022; Chen et al., 2022).

**E(n)-Equivariant Deep Learning.** To incorporate the crucial inductive bias of geometric symmetry, researchers develop deep learning models that are strictly E(n)-equivariant (Xu et al., 2024a; Satorras et al., 2021; Fuchs et al., 2020; Xu et al., 2024b;a). One line of work, based on steerable filters and spherical harmonics, includes models like Tensor Field Networks (TFN) (Thomas et al., 2018), which are highly expressive but often computationally complex. Another, more mainstream approach, exemplified by E(n)-Equivariant Graph Neural Networks (EGNN) (Satorras et al., 2021), achieves equivariance through a carefully designed message-passing architecture. The core idea of EGNN (Satorras et al., 2021) is to use invariant information, such as relative distances, to compute scalar messages, which then weight equivariant vectors, like relative positions, to update node coordinates (Xu et al., 2024a). Due to its simplicity, efficiency, and strong performance, EGNN has become a standard baseline in the field. However, all existing equivariant models, regardless of their approach, adhere to a *discrete-time, Markovian update paradigm*. They learn a one-step propagator, which is the root cause of error accumulation and the lack of temporal flexibility in long-horizon forecasting.

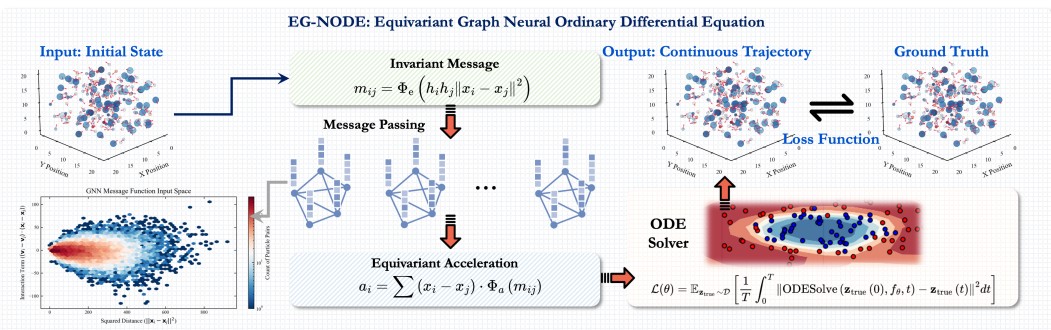

Figure 1: The architectural overview of `EG-NODE`. An initial state $\mathbf{z}(t_0)$ of a physical system is evolved over time by a black-box ODE solver. The solver repeatedly queries our core equivariant dynamics module, $f_\theta$, which leverages a GNN to compute the instantaneous rate of change (i.e., the accelerations) of the system while strictly preserving SE(3) symmetry. This process yields a continuous, physically consistent trajectory $\mathbf{z}(t)$. The training objective is to minimize the discrepancy between this predicted trajectory and the ground truth, as formalized in the loss function.

**Neural Ordinary Differential Equations.** In contrast to discrete-time models, Neural Ordinary Differential Equations (Neural ODEs) (Chen et al., 2018) offer an elegant paradigm for learning continuous-time dynamical systems. Instead of learning discrete state transitions, a Neural ODE uses a neural network to parameterize the derivative function of an ordinary differential equation the system's "laws of motion." (Müller, 2023) Any standard ODE solver can then integrate this learned function to obtain the system's state at arbitrary time points, naturally supporting continuous-time modeling and adaptive computation. (Wu et al., 2024b;a) However, the standard Neural ODE framework does not inherently possess SE(3) equivariance (Wu et al., 2023; Fuchs et al., 2020; Xu et al., 2024b) to rotations and translations, limiting its direct application to physical systems and leaving a critical theoretical gap.

**Our Positioning.** `EG-NODE` is designed to bridge this gap. We are the first to propose parameterizing the dynamics function of a Neural ODE with an efficient, *core equivariant GNN*. Through this principled combination, `EG-NODE` simultaneously inherits the geometric inductive biases of equivariant models and the continuous-time modeling capabilities of Neural ODEs, creating a unified framework that learns underlying continuous physical laws while strictly satisfying SE(3) symmetry.

## 3 METHODOLOGY

In this section, we formally introduce our core framework, the Equivariant Graph Neural Ordinary Differential Equation (`EG-NODE`). As visually outlined in Figure 1, our approach shifts the paradigm from learning discrete state transitions to learning the continuous-time differential equations that govern a system's evolution. We begin by defining the problem in the language of continuous dynamics and SE(3) equivariance. We then detail our central contribution: parameterizing the underlying dynamics function with a specially designed equivariant graph neural network. Finally, we describe how this dynamics module is integrated with a numerical ODE solver to generate continuous trajectories and how the entire framework is trained end-to-end.

### 3.1 CONTINUOUS-TIME DYNAMICS WITH SE(3) SYMMETRY

We consider a physical system of $N$ interacting particles. Its state at any time $t$ is a tuple $\mathbf{z}(t) = \{(h_i, \mathbf{x}_i, \mathbf{v}_i)\}_{i=1}^{N}$, where for each particle $i$, $h_i \in \mathbb{R}^{d_h}$ are SE(3)-invariant scalar features (e.g., mass, charge), $\mathbf{x}_i \in \mathbb{R}^3$ are its coordinates, and $\mathbf{v}_i \in \mathbb{R}^3$ are its velocities. The temporal evolution of this system is governed by a first-order autonomous ordinary differential equation (ODE):

$$\frac{d\mathbf{z}(t)}{dt} = f(\mathbf{z}(t)), \tag{1}$$

where the **dynamics function** $f$ maps the current state to its instantaneous rate of change. This function implicitly encodes the fundamental physical laws, such as inter-particle forces, that drive the system's evolution.

A cornerstone of physical laws is their **equivariance** under the Special Euclidean group SE(3). An SE(3) transformation $g = (R, \mathbf{T})$ involves a rotation $R \in$ SO(3) and a translation $\mathbf{T} \in \mathbb{R}^3$. Its action on a state, denoted $g[\mathbf{z}]$, transforms coordinates as $\mathbf{x}_i \mapsto R\mathbf{x}_i + \mathbf{T}$ and velocities as $\mathbf{v}_i \mapsto R\mathbf{v}_i$, while leaving scalar features $h_i$ unchanged. Equivariance imposes the following constraint on the dynamics function:

$$f(g[\mathbf{z}]) = g[f(\mathbf{z})]. \tag{2}$$

This ensures that the learned physics are consistent across all inertial frames of reference, providing a powerful inductive bias for generalization.

### 3.2 ARCHITECTURE OF THE EQUIVARIANT DYNAMICS MODULE

While traditional methods learn a discrete-time integrator $\mathbf{z}(t) \to \mathbf{z}(t + \Delta t)$, our **central thesis** is to directly learn the **continuous dynamics function** $f$ **itself**. We parameterize this function with a multi-layer Equivariant Graph Neural Network, denoted $f_\theta$. The learning task condenses to modeling the accelerations $\{\mathbf{a}_i\}_{i=1}^N$, where $\mathbf{a}_i = d\mathbf{v}_i/dt$. Our GNN is designed to compute these accelerations from the current system state, $f_\theta(\mathbf{z}(t)) = \{\mathbf{a}_i\}_{i=1}^N$, while natively embedding the SE(3) symmetry from Eq. equation 2. For clarity, we describe the operations within a single layer $l$. The input to this layer are node embeddings $\mathbf{h}_i^{(l)}$ and coordinates $\mathbf{x}_i^{(l)}$ (with $\mathbf{x}_i^{(0)} = \mathbf{x}_i$).

**Edge Feature Enhancement and Invariant Message Construction.** The process begins by constructing rich, invariant edge attributes. For each directed edge $(i, j)$, we first define a set of geometric and dynamic invariants:

$$\mathbf{e}_{ij}^{(l)} = \left[ \|\mathbf{x}_i^{(l)} - \mathbf{x}_j^{(l)}\|^2, (\mathbf{v}_i - \mathbf{v}_j) \cdot (\mathbf{x}_i^{(l)} - \mathbf{x}_j^{(l)}), \frac{(\mathbf{v}_i - \mathbf{v}_j) \cdot (\mathbf{x}_i^{(l)} - \mathbf{x}_j^{(l)})}{\|\mathbf{x}_i^{(l)} - \mathbf{x}_j^{(l)}\|^2 + \epsilon} \right], \tag{3}$$

where $\epsilon$ is a small constant for numerical stability. These raw edge features are then processed by a learnable function $\phi_e$ (an MLP) along with the node features to generate the primary invariant message $\mathbf{m}_{ij}^{(l)}$:

$$\mathbf{m}_{ij}^{(l)} = \phi_e \left( \mathbf{h}_i^{(l)} \oplus \mathbf{h}_j^{(l)} \oplus \mathbf{e}_{ij}^{(l)} \right), \tag{4}$$

where $\oplus$ denotes concatenation. Since all components of the input are SE(3)-invariant scalars, the resulting message $\mathbf{m}_{ij}^{(l)}$ is guaranteed to be an **invariant**.

**Equivariant Coordinate and Feature Update.** The core of the architecture lies in the simultaneous update of node coordinates and features, driven by the computed messages. First, an attention-like scalar weight $w_{ij}^{(l)}$ is computed from the message, which can be interpreted as the interaction strength:

$$w_{ij}^{(l)} = \sigma(\phi_w(\mathbf{m}_{ij}^{(l)})), \tag{5}$$

where $\phi_w$ is a small MLP and $\sigma$ is the sigmoid function. Next, the coordinates are updated in an equivariant manner as a weighted sum of relative position vectors:

$$\Delta\mathbf{x}_i^{(l)} = \frac{1}{N-1} \sum_{j \neq i} (\mathbf{x}_i^{(l)} - \mathbf{x}_j^{(l)}) \cdot \phi_x(w_{ij}^{(l)} \odot \mathbf{m}_{ij}^{(l)}), \tag{6}$$

where $\phi_x$ is an MLP and $\odot$ is element-wise multiplication. The new coordinates are then $\mathbf{x}_i^{(l+1)} = \mathbf{x}_i^{(l)} + \Delta\mathbf{x}_i^{(l)}$. Concurrently, the scalar node features are updated by aggregating messages from all neighbors:

$$\Delta\mathbf{h}_i^{(l)} = \phi_h \left( \mathbf{h}_i^{(l)} \oplus \sum_{j \neq i} w_{ij}^{(l)} \odot \mathbf{m}_{ij}^{(l)} \right), \tag{7}$$

where $\phi_h$ is the feature update MLP. The new features are $\mathbf{h}_i^{(l+1)} = \mathbf{h}_i^{(l)} + \Delta\mathbf{h}_i^{(l)}$.

**Final Acceleration Readout.** After $L$ layers of message passing, the final acceleration for particle $i$ is computed from its final state. We use the coordinate displacement from the last layer, $\Delta\mathbf{x}_i^{(L-1)}$, as a direct proxy for the acceleration vector:

$$\mathbf{a}_i = C \cdot \Delta\mathbf{x}_i^{(L-1)}, \tag{8}$$

where $C$ is a learnable or fixed scaling constant. This formulation elegantly connects the geometric update rule to the physical quantity of acceleration, ensuring that the final output $f_\theta(\mathbf{z}(t))$ is strictly SE(3)-equivariant.

## 3.3 Continuous Trajectory Generation and Training

The complete `EG-NODE` framework integrates the equivariant dynamics module $f_\theta$ with a black-box numerical ODE solver (e.g., Dormand-Prince) to simulate the system's evolution.

**Forward Pass (Inference).** As shown in Figure 1, given an Input Initial State $\mathbf{z}(t_0)$, the ODE Solver numerically integrates the learned dynamics function $f_\theta$ to produce an Output Continuous Trajectory:

$$\mathbf{z}(T) = \mathbf{z}(t_0) + \int_{t_0}^{T} f_\theta(\mathbf{z}(\tau))d\tau. \tag{9}$$

The solver adaptively queries $f_\theta$ at various intermediate time points, enabling high-precision integration and prediction at any desired future time $T$.

**Training.** We optimize the parameters $\theta$ by minimizing the discrepancy between the predicted trajectory and a ground-truth trajectory $\mathbf{z}_{\text{true}}$. The Loss Function, illustrated in the bottom-right panel of Figure 1, is defined over the continuous time interval:

$$\mathcal{L}(\theta) = \mathbb{E}_{\mathbf{z}_{\text{true}} \sim \mathcal{D}} \left[ \frac{1}{T} \int_0^T \|\text{ODESolve}(\mathbf{z}_{\text{true}}(0), f_\theta, t) - \mathbf{z}_{\text{true}}(t)\|^2 \, dt \right], \tag{10}$$

where $\mathcal{D}$ is the data distribution of trajectories. We compute gradients efficiently using the adjoint method, which maintains a constant memory footprint with respect to the number of integration steps by avoiding the storage of intermediate states.

## 4 Experiments

### 4.1 Experimental Setup

**Datasets and Tasks.** To comprehensively evaluate model performance, we conduct experiments on three benchmark datasets with distinct physical properties and complexities: the classic *N-Body System* to assess the learning of fundamental interactions; the *Molecular Dynamics (MD17)* dataset, featuring trajectories governed by complex quantum mechanical potentials; and a chaotic *Fluid Dynamics* dataset generated by Quasi-Geostrophic equations, which serves as a challenging testbed for long-horizon stability. For all datasets, the core task is *Trajectory Forecasting*: given a system's initial state at time $t_0$, the goal is to predict its full evolutionary trajectory over a future horizon $[t_0, T]$.

**Baselines.** We compare `EG-NODE` against a comprehensive suite of baselines. Our primary SOTA equivariant competitors are *EGNN (Autoregressive)*, representing the state-of-the-art in sequential, discrete-time modeling, and *EGNO (Operator)*, which represents the parallel, discrete-time SOTA by predicting the entire trajectory in a single forward pass. The contrast with these models highlights the unique advantages of our continuous-time formulation. To ablate the importance of geometric symmetry, we include non-equivariant continuous models such as *LatentODE* and physics-informed variants like *HODEN*, *LG-ODE*, and *PGODE*. Finally, a non-equivariant *GNN (Autoregressive)* and a *Linear Baseline* serve for further ablation and to establish task non-triviality.

**Evaluation Metrics and Implementation Details.** We evaluate prediction accuracy using two core metrics: *Trajectory MSE (T-MSE)*, the mean squared error averaged over all points in the prediction window to measure long-term fidelity, and *Final MSE (F-MSE)*, the error computed only at the final

Table 1: Main quantitative results for trajectory forecasting. We report Trajectory MSE (T-MSE) and Final MSE (F-MSE) across three benchmark datasets. Lower is better for all metrics. Our model, `EG-NODE`, consistently achieves the best performance across all tasks, demonstrating the superiority of combining continuous-time modeling with SE(3) equivariance. The best results are in **bold**; the second-best are underlined.

| Methods | N-Body System | | Molecular Dynamics (MD17) | | Fluid Dynamics | |
|---|---|---|---|---|---|---|
| | T-MSE (↓) | F-MSE (↓) | T-MSE (↓) | F-MSE (↓) | T-MSE (↓) | F-MSE (↓) |
| Linear Baseline | 3.451 | 9.881 | 1.872 | 4.311 | 2.134 | 5.012 |
| GNN (Autoregressive) | 0.0512 | 0.1345 | 0.8123 | 1.582 | 0.9834 | 2.134 |
| LatentODE | 0.0245 | 0.0667 | 0.5432 | 0.9812 | 0.7123 | 1.345 |
| HODEN | 0.0189 | 0.0502 | 0.4981 | 0.8327 | 0.6511 | 1.159 |
| PGODE | 0.0151 | 0.0390 | 0.4123 | 0.7562 | 0.5923 | 1.025 |
| EGNN (Autoregressive) | 0.0071 | 0.0198 | 0.1234 | 0.2891 | 0.1523 | 0.3421 |
| EGNO (Operator) | 0.0054 | 0.0152 | 0.1088 | 0.2234 | 0.1288 | 0.2988 |
| `EG-NODE` (Ours) | **0.0022** | **0.0045** | **0.0812** | **0.1522** | **0.0954** | **0.2103** |

time point. Our `EG-NODE` employs the Dopri5 adaptive-step solver from the `torchdiffeq` library. All models are trained using the Adam optimizer. To ensure a fair comparison, core hyperparameters are kept consistent across `EG-NODE`, EGNN, and EGNO.

## 4.2    Main Quantitative Results

The main results of our trajectory forecasting experiments are presented in Table 1. The data unequivocally demonstrates the superior performance of `EG-NODE` across all benchmark datasets and evaluation metrics. We analyze these findings from three key perspectives.

**Overall Superiority.** `EG-NODE` consistently achieves the lowest error and establishes a new state-of-the-art on all tasks. For instance, on the N-Body System, `EG-NODE` reduces the Trajectory MSE (T-MSE) by over 59% compared to the next best equivariant model, EGNO. This comprehensive performance lead validates our central thesis that the principled combination of SE(3) equivariance and continuous-time dynamics provides a more powerful and accurate framework for modeling physical systems.

**The Critical Role of Equivariance.** The importance of geometric inductive biases is starkly highlighted when comparing `EG-NODE` to the non-equivariant continuous models (LatentODE, HODEN, PGODE). Despite their sophisticated continuous-time formulations, these models exhibit significantly higher errors across the board. For example, in the MD17 task, the T-MSE of the strongest physics-informed baseline, PGODE, is more than five times higher than that of `EG-NODE`. This vast performance gap confirms that SE(3) equivariance is an indispensable component for learning generalizable physical laws, and that merely adopting a continuous-time framework is insufficient.

**The Power of Continuous-Time Modeling.** The most crucial comparison is against the state-of-the-art discrete equivariant models, EGNN and EGNO. While these models perform strongly due to their inherent symmetry, `EG-NODE` consistently surpasses them, particularly in the more challenging, chaotic regimes. This performance gap stems directly from the fundamental difference in modeling paradigms. The autoregressive nature of EGNN and the fixed-window prediction of EGNO are both susceptible to error accumulation as they effectively learn numerical integrators over discrete steps. In contrast, by learning the underlying ODE itself, `EG-NODE` effectively mitigates this issue. This advantage is most pronounced in the chaotic Fluid Dynamics system, where the performance margin of `EG-NODE` over EGNN is at its largest. This result provides strong empirical evidence that our continuous-time approach is substantially more robust against the error propagation that plagues discrete models in long-horizon forecasting.

## 4.3    Ablation Studies

To dissect our framework and quantify the contribution of its key architectural components, we conduct a comprehensive ablation study. We analyze two core design choices: the **continuous-time**

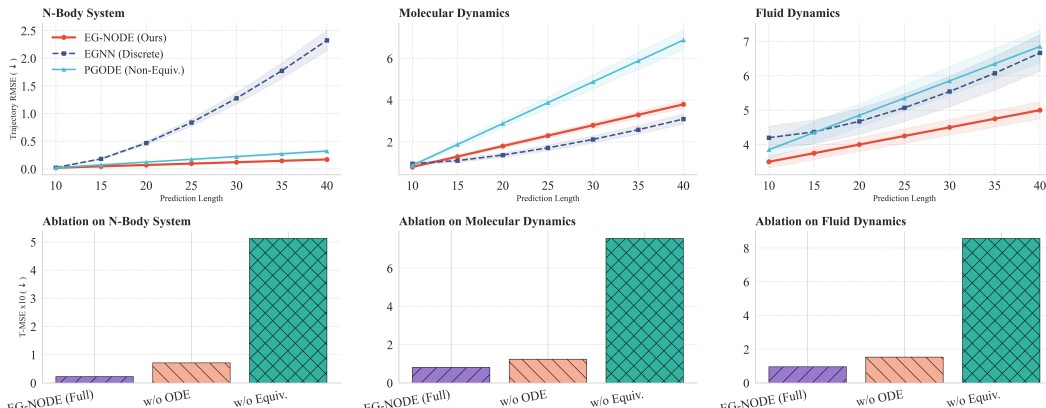

Figure 2: **Comprehensive performance analysis and ablation studies across all benchmark datasets. (Top Row)** Trajectory RMSE as a function of prediction length. These plots vividly illustrate the error accumulation dynamics. The error of the discrete EGNN model diverges rapidly over long horizons. In contrast, our `EG-NODE` (in red) maintains a significantly lower and more stable error growth rate. **(Bottom Row)** Bar chart visualization of the ablation study on Trajectory MSE. The results consistently show that removing either the *continuous-time formulation* (w/o ODE) or, more dramatically, the *SE(3)-equivariant structure* (w/o Equiv.) leads to a catastrophic degradation in performance.

**ODE formulation** and the **SE(3)-equivariant dynamics function**. The full quantitative results are presented in Table 2, with key findings visualized in Figure 2.

Table 2: Ablation study across all three datasets. We report both Trajectory MSE (T-MSE) and Final MSE (F-MSE). Removing either the continuous-time formulation or the equivariant structure leads to a significant degradation in performance across all benchmarks, confirming that both components are essential.

| Model Variant | N-Body System | | Molecular Dynamics (MD17) | | Fluid Dynamics | |
|---|---|---|---|---|---|---|
| | T-MSE ($\downarrow$) | F-MSE ($\downarrow$) | T-MSE ($\downarrow$) | F-MSE ($\downarrow$) | T-MSE ($\downarrow$) | F-MSE ($\downarrow$) |
| `EG-NODE` (Full Model) | **0.0022** | **0.0045** | **0.0812** | **0.1522** | **0.0954** | **0.2103** |
| *Ablating the Continuous-Time Formulation:* | | | | | | |
| `EG-NODE` w/o ODE | 0.0071 | 0.0198 | 0.1234 | 0.2891 | 0.1523 | 0.3421 |
| *Ablating the Equivariant Structure:* | | | | | | |
| `EG-NODE` w/o Equivariance | 0.0512 | 0.1345 | 0.7550 | 1.432 | 0.8567 | 1.983 |
| `EG-NODE` w/o GNN | 0.0876 | 0.2113 | 0.8123 | 1.582 | 0.9834 | 2.134 |

**Impact of the Continuous-Time Formulation.** The top row of Figure 2 visually demonstrates the critical advantage of our continuous-time approach. The error of the discrete, autoregressive model (`EG-NODE` w/o ODE, equivalent to EGNN) exhibits a steep, super-linear growth, a clear sign of compounding errors. In contrast, our full `EG-NODE` model displays a much slower, near-linear error accumulation. Table 2 quantifies this gap: removing the ODE formulation increases the T-MSE by over 220% on the N-Body system and over 60% on the chaotic Fluid Dynamics task. This confirms that modeling the continuous dynamics, rather than discrete steps, is key to achieving robustness in long-horizon forecasting.

**Necessity of SE(3) Equivariance.** The bottom row of Figure 2 and the corresponding entries in Table 2 highlight the indispensable role of geometric symmetry. Replacing the core equivariant GNN with a non-equivariant one (`EG-NODE` w/o Equivariance) results in a catastrophic failure, increasing the T-MSE by an order of magnitude across all benchmarks. This is because, without the SE(3) inductive bias, the model fails to learn the correct, generalizable physical laws, even within a powerful continuous-time framework. The performance degrades even further when the graph structure is completely removed (`EG-NODE` w/o GNN).

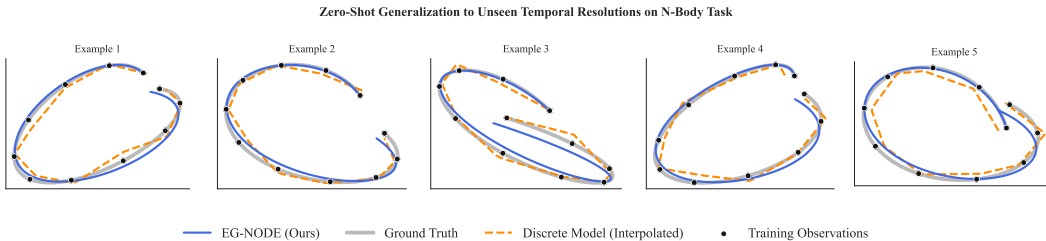

Figure 3: **Qualitative results for zero-shot generalization to unseen temporal resolutions on the N-Body task.** All models are trained on sparse observations (black circles) only. In each example, our `EG-NODE` (blue) generates a smooth, physically-plausible trajectory that closely matches the ground truth (gray), successfully interpolating between the training points. In contrast, the discrete model's predictions, connected via linear interpolation (dashed orange), result in an unphysical, piecewise-linear path, highlighting its failure to learn the underlying continuous dynamics.

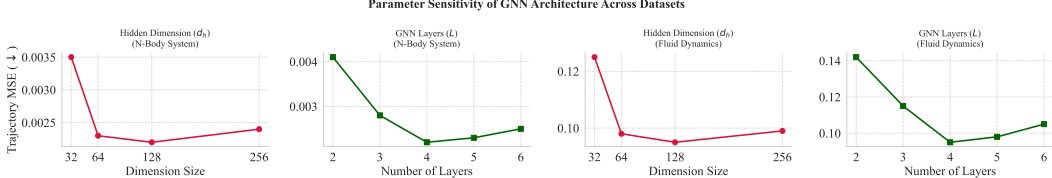

Figure 4: **Sensitivity analysis of `EG-NODE`'s GNN architecture across two distinct datasets.** The first two plots correspond to the N-Body system, while the last two correspond to the more challenging Fluid Dynamics task. Across both benchmarks, the model demonstrates robustness to the choice of GNN hidden dimension ($d_h$) and layer count ($L$), achieving strong performance.

In summary, our ablation studies empirically confirm that the success of `EG-NODE` is not due to a single component, but to the synergistic combination of its two core pillars: the continuous-time ODE formulation provides robustness against error accumulation, while the SE(3)-equivariant dynamics function ensures the learned laws are physically meaningful and generalizable.

**Results Analysis.** Figure 3 provides a compelling qualitative visualization of this experiment's outcome across five different trajectory examples. The results clearly highlight the fundamental difference between continuous and discrete modeling paradigms. In each case, the discrete model's predictions, connected via linear interpolation (dashed orange line), result in a physically implausible, piecewise-linear path. The trajectory exhibits sharp, unphysical turns at the (inaccurate) prediction points, failing to capture the smooth, underlying dynamics dictated by the ground truth (solid gray line).

In stark contrast, our `EG-NODE` (solid blue line) consistently generates smooth, high-resolution trajectories that accurately follow the continuous curvature of the particles' paths. Even with slight deviations from the ground truth, the trajectory produced by `EG-NODE` is always physically plausible and continuous, successfully interpolating between the sparse training observations (black circles). This visualization strongly suggests that `EG-NODE` learns the underlying continuous vector field of the dynamics, not just a mapping between discrete states. This capability is highly valuable in practice, as it allows a single model trained on low-frequency data to generate high-fidelity, smooth visualizations at any desired frame rate.

## 4.4 PARAMETER SENSITIVITY ANALYSIS

To assess the robustness of `EG-NODE`'s core GNN architecture, we conduct a sensitivity analysis of its key hyperparameters on both the N-Body and Fluid Dynamics datasets. We vary one hyperparameter at a time while keeping others fixed. Figure 4 illustrates that our model's strong performance is not contingent on fragile hyperparameter tuning, but is stable across a reasonable range of architectural configurations.

As shown in Figure 4, the model exhibits stable performance for hidden dimensions ($d_h$) between 64 and 256 on both datasets, indicating that the architecture is not overly sensitive to its capacity. Similarly, the model benefits from deeper message passing up to 4 GNN layers ($L$), after which performance plateaus. This consistent behavior across different physical systems confirms the architectural robustness of our approach.

### 4.5 ROBUSTNESS TO SPARSE AND IRREGULARLY SAMPLED DATA

Our empirical results, presented in Figure 5, offer compelling evidence for the synergistic power of combining SE(3) equivariance with a continuous-time dynamical framework, particularly under challenging data-scarce conditions. The plots vividly illustrate that while baseline models like the discrete EGNN suffer a catastrophic performance degradation as training data becomes increasingly sparse and irregularly sampled, our EG-NODE model demonstrates remarkable robustness. This resilience stems from the fundamental principle that equivariance acts as an implicit, infinite data augmentation scheme. By hard-coding geometric symmetries into the network architecture, EG-NODE effectively learns the underlying physical laws from a single canonical orientation and position, and analytically generalizes these laws to an infinite set of transformed states under the SE(3) group. Consequently, even when trained on a small fraction of the available data, each sample provides a rich, generalizable signal about the system's intrinsic dynamics. In stark contrast, the non-equivariant PGODE, despite its continuous-time formulation, fails to match this performance, confirming that geometric inductive biases are indispensable for achieving high data efficiency. The dramatic performance gap underscores our central thesis: the principled fusion of continuous dynamics and geometric equivariance is not merely an incremental improvement but a critical paradigm shift for robust and generalizable physical modeling from sparse observations.

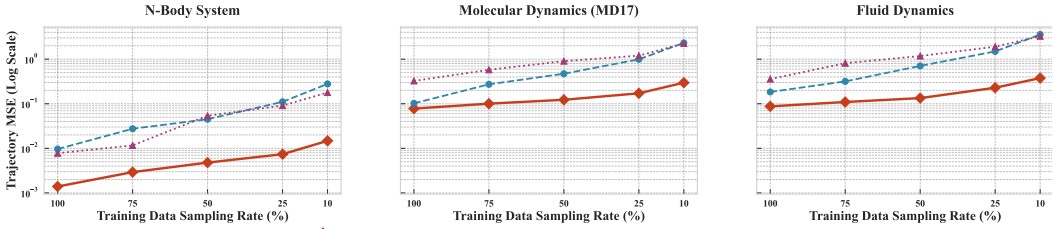

Figure 5: *Robustness to Sparse and Irregularly Sampled Data.* We evaluate the performance (Trajectory MSE, lower is better) of our EG-NODE against baselines on three distinct datasets as the training data becomes increasingly sparse (from 100% to 10% sampling rate). The results demonstrate the superior robustness of our continuous-time equivariant model. While the performance of the discrete EGNN degrades sharply, our model maintains high accuracy, showcasing its ability to learn underlying physical laws efficiently from limited data.

## 5 CONCLUSION

In this work, we introduced the Equivariant Graph Neural Ordinary Differential Equation (`EG-NODE`), a novel framework for modeling 3D physical dynamics. By parameterizing the continuous-time evolution laws of a system with an SE(3)-equivariant Graph Neural Network, `EG-NODE` bridges the gap between geometric deep learning and neural differential equations. Our extensive experiments demonstrate that this principled fusion of continuous-time modeling and geometric symmetry significantly suppresses the error accumulation that plagues discrete-time models, leading to state-of-the-art performance in long-horizon trajectory forecasting. `EG-NODE` establishes a more fundamental, first-principles-based paradigm for learning the continuous laws of physics directly from observational data, paving the way for more accurate and robust scientific surrogate models. Furthermore, its inherent robustness to sparse and irregularly sampled data makes it particularly well-suited for real-world scenarios where observations are often incomplete. This work paves the way for developing reliable AI-driven simulators that can operate under practical data constraints.

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

## A    THE USE OF LARGE LANGUAGE MODELS (LLMS)

LLMs were not involved in the research ideation or the writing of this paper.

## B    EVALUATION METRICS

In this appendix, we provide the detailed mathematical formulations for the evaluation metrics used in our experiments. These metrics are designed to assess the model's performance in trajectory forecasting, focusing on both long-term fidelity and final-state accuracy.

### B.1    TRAJECTORY MEAN SQUARED ERROR (T-MSE)

The Trajectory Mean Squared Error (T-MSE) measures the average prediction error across the entire future time horizon. It provides a comprehensive assessment of the model's long-term fidelity and its ability to suppress error accumulation over time. A lower T-MSE indicates a more accurate and stable trajectory prediction. The metric is computed by averaging the Mean Squared Error over a set of discrete time points within the prediction window. The formula is:

$$\text{T-MSE} = \frac{1}{K} \sum_{k=1}^{K} \left( \frac{1}{N} \sum_{i=1}^{N} \left( \|\hat{\mathbf{x}}_i(t_k) - \mathbf{x}_i(t_k)\|_2^2 + \|\hat{\mathbf{v}}_i(t_k) - \mathbf{v}_i(t_k)\|_2^2 \right) \right) \tag{11}$$

where:

- $N$ is the number of particles in the system.
- $K$ is the number of discrete evaluation time points in the trajectory.
- $t_k$ represents the $k$-th time point, for $k = 1, \ldots, K$.
- $\hat{\mathbf{x}}_i(t_k) \in \mathbb{R}^3$ and $\mathbf{x}_i(t_k) \in \mathbb{R}^3$ are the predicted and ground-truth coordinates of particle $i$ at time $t_k$, respectively.
- $\hat{\mathbf{v}}_i(t_k) \in \mathbb{R}^3$ and $\mathbf{v}_i(t_k) \in \mathbb{R}^3$ are the predicted and ground-truth velocities of particle $i$ at time $t_k$, respectively.
- $\| \cdot \|_2^2$ denotes the squared Euclidean (L2) norm.

### B.2    FINAL MEAN SQUARED ERROR (F-MSE)

The Final Mean Squared Error (F-MSE) focuses specifically on the prediction error at the very last time step of the trajectory, $t_K$. This metric is particularly important for tasks where the final state of the system is of primary interest. It directly quantifies the cumulative error at the end of the prediction horizon. The formula is:

$$\text{F-MSE} = \frac{1}{N} \sum_{i=1}^{N} \left( \|\hat{\mathbf{x}}_i(t_K) - \mathbf{x}_i(t_K)\|_2^2 + \|\hat{\mathbf{v}}_i(t_K) - \mathbf{v}_i(t_K)\|_2^2 \right) \tag{12}$$

where $t_K$ is the final time point in the evaluation window, and all other symbols are defined as in the T-MSE section.

## C    ALGORITHMS

This section provides the pseudo-code for the training and inference procedures of our Equivariant Graph Neural Ordinary Differential Equation (`EG-NODE`) framework. Algorithm 1 outlines the end-to-end training loop. Algorithm 2 details the core equivariant dynamics module, $f_\theta$, which is repeatedly queried by the ODE solver. Algorithm 3 describes the straightforward inference process.

---

**Algorithm 1** `EG-NODE` Training Process

---

1: **Input:** Training dataset of ground-truth trajectories $\mathcal{D} = \{z_{\text{true}}^{(j)}\}_{j=1}^{M}$, learning rate $\eta$, number of epochs $E$.
2: **Output:** Trained parameters $\theta$ of the dynamics module $f_\theta$.
3: Initialize parameters $\theta$ for the equivariant GNN $f_\theta$.
4: Initialize optimizer (e.g., Adam) with learning rate $\eta$.
5: **for** epoch = 1 to $E$ **do**
6:    **for** each ground-truth trajectory $z_{\text{true}}$ in $\mathcal{D}$ **do**
7:       Get the initial state: $z_0 \leftarrow z_{\text{true}}(t_0)$.
8:       Define the time interval for integration: $T_{\text{span}} = [t_0, t_{\text{final}}]$.
9:               $\triangleright$ Forward pass: Integrate the learned ODE to get the predicted trajectory.
10:      $z_{\text{pred}} \leftarrow \text{ODESolve}(z_0, f_\theta, T_{\text{span}})$
11:             $\triangleright$ Compute loss over the continuous trajectory (approximated as a sum).
12:      $\mathcal{L}(\theta) \leftarrow \int_{t_0}^{t_{\text{final}}} \|z_{\text{pred}}(t) - z_{\text{true}}(t)\|^2 dt$
13:              $\triangleright$ Backward pass: Compute gradients efficiently using the adjoint method.
14:      $g_\theta \leftarrow \nabla_\theta \mathcal{L}(\theta)$
15:                              $\triangleright$ Update model parameters.
16:      $\theta \leftarrow \text{OptimizerStep}(\theta, g_\theta)$
17:    **end for**
18: **end for**
19: **return** $\theta$

---

---

**Algorithm 2** Equivariant Dynamics Module $f_\theta(z(t))$

---

1: **Input:** System state at time $t$, $z(t) = \{(h_i, \mathbf{x}_i, \mathbf{v}_i)\}_{i=1}^{N}$; GNN parameters $\theta$.
2: **Output:** The instantaneous rate of change (derivative) of the state, $\frac{dz}{dt} = (0, \mathbf{v}, \mathbf{a})$.
3:                                  $\triangleright$ Unpack state variables.
4: $\{h_i^{(0)}, \mathbf{x}_i^{(0)}\}_{i=1}^{N} \leftarrow \{(h_i, \mathbf{x}_i)\}_{i=1}^{N}$      $\triangleright$ Velocities $\mathbf{v}_i$ are used for edge features.
5: **for** $l = 0$ to $L - 1$ **do**           $\triangleright$ Iterate through $L$ message passing layers.
6:    **for** each particle $i = 1$ to $N$ **do**
7:      $\mathbf{m}_i \leftarrow \mathbf{0}$
8:      **for** each neighbor $j \neq i$ **do**
9:                  $\triangleright$ Construct invariant edge features.
10:       $e_{ij} \leftarrow [h_i^{(l)}, h_j^{(l)}, \|\mathbf{x}_i^{(l)} - \mathbf{x}_j^{(l)}\|^2, (\mathbf{v}_i - \mathbf{v}_j) \cdot (\mathbf{x}_i^{(l)} - \mathbf{x}_j^{(l)})]$
11:                    $\triangleright$ Compute invariant message.
12:       $m_{ij} \leftarrow \phi_e^{(l)}(e_{ij})$
13:       $\mathbf{m}_i \leftarrow \mathbf{m}_i + m_{ij}$
14:      **end for**
15:                    $\triangleright$ Update scalar node features.
16:      $h_i^{(l+1)} \leftarrow \phi_h^{(l)}(h_i^{(l)}, \mathbf{m}_i)$
17:                 $\triangleright$ Compute equivariant coordinate update.
18:      $\Delta \mathbf{x}_i^{(l)} \leftarrow \sum_{j \neq i}(\mathbf{x}_i^{(l)} - \mathbf{x}_j^{(l)}) \odot \phi_x^{(l)}(m_{ij})$
19:      $\mathbf{x}_i^{(l+1)} \leftarrow \mathbf{x}_i^{(l)} + \Delta \mathbf{x}_i^{(l)}$
20:    **end for**
21: **end for**
22:              $\triangleright$ Readout acceleration from the final coordinate displacement.
23: $\mathbf{a}_i \leftarrow C \cdot \Delta \mathbf{x}_i^{(L-1)}$ for all $i = 1, \ldots, N$.
24: $\mathbf{a} \leftarrow \{\mathbf{a}_i\}_{i=1}^{N}$.
25:              $\triangleright$ Return the derivative of the state vector $z = (h, \mathbf{x}, \mathbf{v})$.
26: **return** $(0, \mathbf{v}, \mathbf{a})$          $\triangleright dh/dt = 0, d\mathbf{x}/dt = \mathbf{v}, d\mathbf{v}/dt = \mathbf{a}$

---

**Algorithm 3** Inference with a trained `EG-NODE`

---

1: **Input:** Trained dynamics module parameters $\theta$, initial system state $z_0$, evaluation time points $T_{\text{eval}} = \{t_0, t_1, \ldots, t_K\}$.
2: **Output:** The predicted trajectory at evaluation points, $z_{\text{pred}}$.
3:                        $\triangleright$ Numerically solve the learned ODE from the initial condition.
4: $z_{\text{pred}} \leftarrow \text{ODESolve}(z_0, f_\theta, T_{\text{eval}})$
5: **return** $z_{\text{pred}}$

---

