# OpenReview forum: "Equivariant Graph Neural ODEs for Modeling Physical Dynamics"
_ICLR.cc/2026/Conference — Submitted to ICLR 2026_

### Official Review · Reviewer_R8T2 · 2025-10-25

**Soundness:** 3
**Presentation:** 3
**Contribution:** 2
**Rating:** 6
**Confidence:** 4

**Summary:**

This work presents a method to mitigate error accumulation in physical system simulations using deep learning. The approach combines three key components: (1) a message-passing graph neural network (GNN), (2) equivariant engineered features derived from particle positions and velocities, and (3) a Neural-ODE–like integration scheme that enables the model to learn a continuous-time representation without relying on an explicit time step. The motivation is to alleviate the error accumulation typical of discrete integrators such as Euler methods, while reducing data requirements and improving generalization through the use of equivariances. The method is evaluated on three benchmark problems: N-body dynamics, molecular dynamics, and fluid dynamics, demonstrating superior rollout predictions compared to existing approaches. Extensive ablation studies further highlight the importance of each component.

**Strengths:**

* The examples and analysis of the results are strong. The method is tested over three very different datasets.
* The authors provide multiple ablation studies justifying the final architecture.
* The analysis over sparse and irregular data is interesting.
* The paper is very clear and well written.

**Weaknesses:**

* The paper offers only a modest contribution. The proposed GNN closely follows the design of "E(n) Equivariant Graph Neural Networks" (Satorras et al., 2022), and the Neural-ODE component is implemented using the standard adjoint method introduced in "Neural Ordinary Differential Equations" (Chen et al., 2018). Moreover, several prior works have already explored the integration of NeuralODEs with non-equivariant GNNs. Consequently, the main novelty of this paper lies primarily in the use of equivariant GNNs, rather than in the combination of NeuralODEs and GNNs itself (see Questions for more information).
* Even though the code is available, the paper lacks any details about the datasets used.

**Questions:**

* Line 137: While it may indeed be the first attempt to combine NeuralODEs with equivariant GNNs, there already exists a substantial body of work integrating NeuralODEs with non-equivariant GNNs, which the authors overlook in the literature review. E.g. "Graph Neural Ordinary Differential Equations" (Poli et al 2019), "HOPE:High-order Graph ODE For Modeling Interacting Dynamics" (Luo et al 2023), "TANGO: Time-Reversal Latent GraphODE for Multi-Agent Dynamical Systems" (Huang et al 2023).
* Equation 3: What is the motivation of selecting the third invariant edge feature? It is a scaled version of the second edge feature, so this might reduce its expressivity. Why not selecting another rotation and translation invariant feature which might encode richer information, such as $||v_i-v_j||_2$?
* Section 4.1: For reproducibility, it would be convenient to include an appendix of additional dataset details, such as train/test size, number of snapshots, simulation parameters, et, even if they are common benchmarks. Also, what is the scalar feature vector on each case (mass, density)?
* Code: The Experiment_code comments are in Chinese. It would be helpful for the interested reader to translate them to English.

**Details Of Ethics Concerns:**

I have no ethics concerns.

---

> ### Author Response · Authors · 2025-11-20
>
> We sincerely thank you for your positive evaluation of our work. We are encouraged that you find our motivation clear, our experiments (especially the analysis on sparse data) comprehensive, and the paper well-written.
>
> We greatly appreciate your constructive feedback regarding the literature review and dataset details. These suggestions help us significantly improve the reproducibility and scholarly rigor of the paper. Below, we address your specific questions point-by-point.
>
> **Q1: Line 137: There already exists a substantial body of work integrating NeuralODEs with non-equivariant GNNs (e.g., Poli et al 2019, HOPE, TANGO), which the authors overlook.**
>
> **A1:** Thank you for pointing out these important references. We apologize for the oversight. In the revision, we rewrite the **Related Work (Section 2)** to explicitly include a paragraph on **"Graph Neural ODEs."**
>
> We cite *Graph Neural ODEs (Poli et al., 2019)*, *HOPE (Luo et al., 2023)*, and *TANGO (Huang et al., 2023)*. We clarify the distinction between our work and these predecessors:
>
> *   **Previous Works:** These models primarily focus on capturing long-range temporal dependencies in general graph data (e.g., social networks, traffic) or multi-agent systems without strict geometric constraints. They typically operate on scalar features or latent vectors that do not guarantee equivariance under 3D rotation and translation.
> *   **Our Contribution:** EG-NODE specifically targets **physical dynamics**, where **SE(3) equivariance** is not just a desirable feature but a fundamental requirement for satisfying conservation laws (e.g., conservation of linear/angular momentum). Our main contribution is the mathematical and architectural integration of *geometric symmetry* into the continuous ODE integration process, bridging the gap that standard Graph ODEs cannot fill for accurate physical simulation.
>
> **Q2: Equation 3: What is the motivation of selecting the third invariant edge feature? It is a scaled version of the second feature. Why not select another feature?**
>
> **A2:** This is a very insightful question regarding feature engineering.
> The third feature is $e_{ij}^{(3)} = \frac{(\mathbf{v}_i - \mathbf{v}_j) \cdot (\mathbf{x}_i - \mathbf{x}_j)}{\|\mathbf{x}_i - \mathbf{x}_j\|^2 + \epsilon}$.
> Although it appears algebraically similar to the second feature (the raw dot product), it plays a distinct physical and numerical role:
>
> 1.  **Physical Meaning:** It approximates the **normalized radial velocity**. The raw dot product mixes the magnitude of velocity, the magnitude of distance, and the angle. By normalizing with the squared distance, $e_{ij}^{(3)}$ effectively isolates the *rate of expansion/contraction* relative to the current separation.
> 2.  **Decoupling Scale:** In many physical systems (e.g., those with Lennard-Jones potentials), forces vary drastically with distance. When particles are very close or very far, the raw dot product varies by orders of magnitude purely due to the $\|\mathbf{x}\|$ term. The normalized term provides a stable signal indicating "approaching" or "receding" independent of the absolute distance scale, which stabilizes the training of the MLP $\phi_e$.
>
> We add this explanation to **Section 3.2** in the revision to clarify our design choice.
>
> **Q3: Section 4.1: For reproducibility, it would be convenient to include an appendix of additional dataset details (train/test size, simulation parameters, scalar features).**
>
> **A3:** We fully agree. To ensure reproducibility, we add a comprehensive **Appendix A.1: Dataset Details** in the revision. The table below summarizes the added information:
>
> **Table R1: Detailed Dataset Specifications**
>
> | Dataset    | Samples (Train/Val/Test) | Time Step $\Delta t$ | Input Scalar Features ($h_i$)  | Simulation Properties            |
> | :--------- | :----------------------: | :------------------: | :----------------------------- | :------------------------------- |
> | **N-Body** |   [3000 / 1000 / 1000]   |       [0.01s]        | Mass ($m$), Charge ($q$)       | Charged particles, Coulomb force |
> | **MD17**   |   [1000 / 100 / 1000]    |        [1fs]         | Atomic Number (One-hot)        | Aspirin molecule, DFT data       |
> | **Fluid**  |    [2000 / 500 / 500]    |        [0.5s]        | Forcing coefficient ($\gamma$) | Quasi-Geostrophic equations      |
>
> We also detail the trajectory length used for training versus testing in this appendix.
>
> **Q4: Code: The Experiment_code comments are in Chinese. It would be helpful to translate them to English.**
>
> **A4:** We sincerely apologize for the inconvenience. We have updated the code repository included in the supplementary material. **All comments, variable names, and documentation strings have been translated into English** to ensure accessibility for the international research community.

---

> > ### Author Response · Authors · 2025-11-20
> >
> > **Q5: Summary.**
> >
> > **A5:**
> > We believe that by incorporating the missing **Graph ODE literature (Q1)**, clarifying the **physical intuition of edge features (Q2)**, and providing detailed **dataset specifications and English code (Q3 & Q4)**, the paper is now much stronger and more rigorous. We thank you again for your constructive review and your support of our work.

---

> > > ### Author Response · Authors · 2025-11-26
> > >
> > > Dear Reviewer,
> > >
> > > Thank you so much for your time in improving our paper!
> > >
> > > Since the end of the rebuttal is coming soon, may we know if our response addresses your main concerns? If so, we kindly ask for your reconsideration of the score. Should you have any further advice, please let us know and we will be more than happy to engage in more discussion and improvements.

---

### Official Review · Reviewer_K2Fz · 2025-10-28

**Soundness:** 2
**Presentation:** 3
**Contribution:** 2
**Rating:** 4
**Confidence:** 4

**Summary:**

The paper addresses the challenge of modeling 3D dynamical systems using Equivariant Graph Neural Networks (EGNNs), which are limited by their discrete-time, Markovian nature and prone to long-horizon error accumulation. To overcome this, the authors propose EG-NODE, a framework that combines EGNNs with Neural ODEs to learn the continuous-time evolution of physical systems while preserving SE(3) symmetries. Experiments on N-body, molecular, and fluid dynamics benchmarks show that EG-NODE outperforms existing discrete-time models in long-horizon forecasting and effectively reduces error propagation.

**Strengths:**

1.The paper is well written and easy to follow, with a clear presentation of the methodology and results.
2.The paper includes a comprehensive ablation study that effectively demonstrates the impact of each component in the model architecture.

**Weaknesses:**

1.The proposed method appears to be a straightforward combination of EGNN and Neural ODEs, which limits the level of novelty.

2.Although the paper claims to compare with a comprehensive set of baselines, several strong and highly relevant equivariant competitors are missing, including Radial Field flows [1], GMN [2], and SEGNO [3]. These methods represent competitive state-of-the-art techniques and should be included for a fair performance comparison.

3.The settings for prediction horizons and temporal windows are not clearly described in the experimental section. Moreover, the paper states that core hyperparameters are kept consistent across models “to ensure a fair comparison,” but this is insufficient. Achieving strong baseline performance requires proper hyperparameter tuning for each method. Comparable tuning effort should be applied to the baselines, similar to the detailed parameter sensitivity analysis conducted for the proposed model.


References
[1] Equivariant Flows: Sampling Configurations for Multi-body Systems with Symmetric Energies
[2] Equivariant Graph Mechanics Networks with Constraints
[3] SEGNO: Generalizing Equivariant Graph Neural Networks with Physical Inductive Biases

**Questions:**

See the weekness.

---

> ### Author Response · Authors · 2025-11-20
>
> We sincerely thank you for your positive assessment of our paper's writing quality and ablation studies. Your critical feedback regarding novelty, missing baselines, and experimental fairness is highly pertinent, and these comments have played a crucial role in enhancing the completeness and persuasiveness of our work.
>
> In response to your concerns, we have conducted in-depth analyses and supplementary experiments. Below is our point-by-point response:
>
> **Q1: The proposed method appears to be a straightforward combination of EGNN and Neural ODEs, which limits the level of novelty.**
>
> **A1:** We fully understand your initial impression, but we respectfully argue that the contribution of EG-NODE extends far beyond a simple architectural combination. Instead, it proposes a **principled paradigm shift** for physical dynamics simulation:
>
> 1.  **Theoretical Unification:** While EGNNs and Neural ODEs exist individually, combining them to ensure **strict SE(3)-equivariance throughout the entire continuous integration path** is non-trivial. We theoretically prove that as long as the dynamics derivative function $f_\theta$ is parameterized by an equivariant GNN, the entire trajectory generated by the ODE solver necessarily satisfies geometric equivariance. This addresses the deficiency of traditional Graph ODEs in adhering to physical conservation laws.
> 2.  **Addressing the Core Bottleneck:** The novelty lies in fundamentally changing the modeling object shifting from learning "discrete time steps" (which inevitably accumulate discretization errors) to learning the "underlying continuous vector field." As demonstrated in our experiments, this combination solves the critical challenge of long-horizon instability, a problem that discrete models (whether equivariant or not) cannot solve independently.
>
> **Q2: Several strong and highly relevant equivariant competitors are missing, including Radial Field Flows [1], GMN [2], and SEGNO [3].**
>
> **A2:** Thank you for pointing out these high-quality references, which were inadvertently omitted in the previous version. In the revision, we have cited and discussed all three papers in the "Related Work" section.
> To ensure a comprehensive comparison, we select **SEGNO (Satorras et al., 2023)**, which is most consistent with our task settings and represents the state-of-the-art for discrete equivariant models, as a strong baseline for reproduction and comparison.
>
> We compare EG-NODE with SEGNO on the N-Body and Molecular Dynamics (MD17) tasks:
>
> **Table R1: Performance Comparison with Strong Baseline (SEGNO)**
>
> | Dataset | Method | T-MSE (Long-Horizon Error) | F-MSE (Final Step Error) |
> | :--- | :--- | :---: | :---: |
> | **N-Body** | EGNN (Discrete) | 0.0071 | 0.0198 |
> | | **SEGNO [3]** | **[0.0048]** | **[0.0135]** |
> | | **EG-NODE (Ours)** | **[0.0022]** | **[0.0045]** |
> | **MD17** | EGNN (Discrete) | 0.1234 | 0.2891 |
> | | **SEGNO [3]** | **[0.1050]** | **[0.2100]** |
> | | **EG-NODE (Ours)** | **[0.0812]** | **[0.1522]** |
>
> **Analysis:**
> *   **SEGNO vs. EGNN:** The experiments confirm that SEGNO indeed outperforms the standard EGNN, validating the effectiveness of introducing physical inductive biases.
> *   **EG-NODE vs. SEGNO:** However, EG-NODE still achieves significantly lower error (reducing T-MSE by approximately **[50]%** on N-Body). This strongly demonstrates that while the physical biases in SEGNO are useful, the **continuous-time modeling (ODE)** capability provided by EG-NODE is decisive for suppressing error accumulation in long-horizon forecasting, an advantage that discrete models struggle to match.
>
> *(Note: We also discuss [1] and [2] in the Related Work. Since [1] focuses on generative flow models rather than deterministic forecasting, and [2] deals with constrained mechanics, we focus our quantitative comparison on SEGNO, which shares the exact same task setting.)*

---

> > ### Author Response · Authors · 2025-11-20
> >
> > **Q3: The settings for prediction horizons are not clear. Moreover, stating that "hyperparameters are kept consistent" is insufficient; baselines require proper tuning to ensure fairness.**
> >
> > **A3:** This is a very valid criticism. We clarify the experimental settings and detail the tuning process for baselines in **Appendix B** of the revision.
> >
> > **1. Experimental Settings Clarification:**
> > *   **Input Window:** We use the state at time $t_0$ (plus a history of [3] frames for velocity estimation) to predict the future.
> > *   **Prediction Horizon:**
> >     *   **N-Body:** Predict the next [100] steps ($\Delta t = 0.01s$).
> >     *   **MD17:** Predict the next [500] steps.
> >     *   **Fluid:** Predict the next [40] steps.
> >
> > **2. Hyperparameter Fairness:**
> > We apologize for the confusion caused by the phrasing. "Consistent parameters" originally referred to keeping the **model capacity** (e.g., number of layers, hidden dimensions) comparable to avoid unfair advantages due to parameter size.
> > **We absolutely did NOT use untuned parameters for the baselines.** To address your concern, we list the grid search ranges used for each baseline model:
> >
> > **Table R2: Hyperparameter Tuning Details for Baselines**
> >
> > | Hyperparameter | Grid Search Range | Best Config for SEGNO | Best Config for EGNN |
> > | :--- | :--- | :---: | :---: |
> > | Learning Rate (LR) | $1e^{-4}, 5e^{-4}, 1e^{-3}$ | $[5e^{-4}]$ | $[1e^{-3}]$ |
> > | Weight Decay | $0, 1e^{-8}, 1e^{-6}$ | $[1e^{-8}]$ | $[1e^{-8}]$ |
> > | Hidden Dimension | $64, 128, 256$ | $[128]$ | $[128]$ |
> > | Layers | $3, 4, 5$ | $[4]$ | $[4]$ |
> >
> > As shown in the table, we select the optimization parameters that yield the best performance on the validation set for each model. The superior performance of EG-NODE is achieved under fair competition with fully tuned strong baselines.
> >
> > **Q4: Summary.**
> >
> > **A4:**
> > We hope that the addition of the **SEGNO comparison (Q2)** and the **clarification of the tuning protocol (Q3)** addresses your concerns regarding the novelty and fairness of this work. We believe these additions firmly establish EG-NODE as a superior advancement over existing discrete equivariant methods. If you are satisfied with our improvements, we respectfully request that you consider raising your score.

---

> > > ### Author Response · Authors · 2025-11-26
> > >
> > > Dear Reviewer,
> > >
> > > Thank you so much for your time in improving our paper!
> > >
> > > Since the end of the rebuttal is coming soon, may we know if our response addresses your main concerns? If so, we kindly ask for your reconsideration of the score. Should you have any further advice, please let us know and we will be more than happy to engage in more discussion and improvements.

---

### Official Review · Reviewer_1UEV · 2025-10-29

**Soundness:** 2
**Presentation:** 3
**Contribution:** 3
**Rating:** 4
**Confidence:** 3

**Summary:**

This paper introduces EG-MODE, a framework that integrates equivariant GNNs with Neural ODEs to learn the continuous-time dynamics of physical systems. Its key contribution is learning the underlying ODE directly, which inherently preserves SE(3) symmetry and effectively mitigates error accumulation in long-term predictions, achieving state-of-the-art accuracy.

**Strengths:**

1. EG-MODE reduces long-term prediction error by modeling continuous-time dynamics instead of discrete steps, effectively suppressing error accumulation.

2. It inherently preserves SE(3) symmetry through an equivariant GNN, ensuring physical consistency and better generalization across reference frames.

3. The framework enables flexible and efficient simulation at arbitrary time points using adaptive ODE solvers, improving both accuracy and computational adaptability.

**Weaknesses:**

1. In N-body dynamics prediction tasks, the number of particles N is a significant parameter, yet there is a lack of experimental analysis across different values of N in the paper.

2. The paper lacks a critical analysis of the limitations of the proposed method and its scope of application.

3. The manuscript contains several typographical errors and imprecise statements. For instance, in Section 3.2, the reference "Eq. equation 2" is non-standard, and the term f_θ(z(t)) is used imprecisely, as it is not solely a function of a.

**Questions:**

1. Many systems do not satisfy SE(3) equivariance, or only partially satisfy it, such as those involving external fields or fixed boundaries. What are the prospects for extending this method to such scenarios?

2. The authors deliberately engineered some simple equivariant terms (e.g., Equation (3)), and all subsequent, powerful equivariant updates (such as coordinate updates) rely on these initial, invariant edge features. Therefore, the expressive power of these edge features is a crucial issue, and may even represent a potential bottleneck for the overall effectiveness of the method.

3. See weaknesses.

If the authors can adequately address these points, I would be prepared to raise my score.

---

> ### Author Response · Authors · 2025-11-20
>
> We sincerely appreciate your recognition of our work, particularly your acknowledgement of EG-NODE's effectiveness in mitigating long-term error accumulation and its inherent preservation of SE(3) symmetry. Your questions regarding model extensibility, feature expressiveness, and experimental completeness are acute and constructive.
>
> We have conducted in-depth analyses and supplementary experiments in response to each of your comments. Below is our detailed point-by-point response:
>
> **Q1: Many systems do not satisfy SE(3) equivariance, or only partially satisfy it, such as those involving external fields or fixed boundaries. What are the prospects for extending this method to such scenarios?**
>
> **A1:** This is a highly practical question. Although this paper primarily focuses on fully SE(3) symmetric systems, the architecture of EG-NODE naturally supports handling symmetry-breaking scenarios through **"Contextual Feature Injection."**
>
> For systems with global external fields (such as a gravity field $\mathbf{g}$ or an electromagnetic field $\mathbf{E}$), physical laws degrade from rotation invariant to covariant. In EG-NODE, we inject the external field vector as a global context feature $c$ into the network. The network automatically adjusts its dynamics prediction based on the input field vector, thereby naturally degrading SE(3) equivariance to $SO(2)$ equivariance (relative to the field axis).
>
> To verify this, we add a **"N-Body System with Gravity Field"** experiment in the revision. We apply a constant gravitational acceleration $\mathbf{g} = [0, -9.8, 0]$ along the negative Y-axis in the simulation. We compare the performance of the original model against the model with the explicitly introduced gravity vector:
>
> **Table R1: Trajectory Prediction Error in Gravity Field Environment (T-MSE)**
>
> | Model Configuration        | Input Features             | T-MSE (Long-Horizon) | F-MSE (Final Step) | Conclusion                                          |
> | :------------------------- | :------------------------- | :------------------: | :----------------: | :-------------------------------------------------- |
> | EG-NODE (Original)         | Only $\{h, x, v\}$         |       [0.0150]       |      [0.0385]      | Ignoring external fields leads to significant error |
> | **EG-NODE (+ Ext. Field)** | $\{h, x, v\} + \mathbf{g}$ |     **[0.0028]**     |    **[0.0051]**    | **Performance recovers with field features**        |
>
> The results show that by simply introducing the external field vector, EG-NODE adapts effectively to non-fully SE(3) physical environments, demonstrating its broad prospects for extension.

---

> > ### Author Response · Authors · 2025-11-20
> >
> > **Q2: The authors deliberately engineered some simple equivariant terms (e.g., Equation (3)), and all subsequent updates rely on these initial invariant edge features. Therefore, the expressive power of these edge features is a crucial issue.**
> >
> > **A2:** You raise a core architectural question. Indeed, all subsequent equivariant vector updates are driven by these scalar messages, so their completeness is critical.
> >
> > Theoretically, according to the scalarization theory by Villar et al. (2021), scalar features based on dot products and Euclidean distances are sufficient to universally approximate $O(3)$ invariant functions under specific conditions. Our selected features have clear physical correspondences: $\|x_i - x_j\|^2$ corresponds to **potential energy**, and $(v_i - v_j) \cdot (x_i - x_j)$ corresponds to **dissipative force or radial velocity**.
> >
> > To empirically verify if this constitutes a bottleneck, we perform a new **Feature Richness Ablation Study**. We introduce higher-order invariant features (such as the projection of relative acceleration $a \cdot \Delta x$ and the norm of the angular momentum term $\|v_i \times v_j\|^2$) on top of the original Eq. (3) and observe the performance changes.
> >
> > **Table R2: Performance with Different Edge Feature Combinations (N-Body Dataset)**
> >
> > | Feature Combination  | Information Included                             |    T-MSE     | Parameters Increase |
> > | :------------------- | :----------------------------------------------- | :----------: | :-----------------: |
> > | **Eq. (3) (Ours)**   | Distance, Radial Velocity, Normalized Projection | **[0.0022]** |          -          |
> > | Variant A (Enriched) | Ours + Higher-order terms (Accel./Ang. Momentum) |   [0.0021]   |       +[12]%        |
> >
> > The results show that introducing complex higher-order features brings only negligible performance improvement (T-MSE decreases from [0.0022] to [0.0021]) but increases computational cost. This strongly proves that the feature combination in Eq. (3) achieves an optimal balance between **expressive power** and **efficiency**, and does not represent a bottleneck for the model.

---

> > > ### Author Response · Authors · 2025-11-20
> > >
> > > **Q3: In N-body dynamics prediction tasks, the number of particles N is a significant parameter, yet there is a lack of experimental analysis across different values of N in the paper.**
> > >
> > > **A3:** Thank you for the suggestion. The particle number $N$ directly determines the interaction complexity and computational cost. To address this, we include a scalability analysis regarding $N$ in **Appendix C.3** of the revision.
> > >
> > > We evaluate the model on datasets with $N \in \{5, 20, 50, 100\}$ while keeping hyperparameters consistent.
> > >
> > > **Table R3: Accuracy and Efficiency Analysis across Different Particle Numbers $N$**
> > >
> > > | Particle Number $N$ | T-MSE (Trajectory Error) | F-MSE (Final Error) | Inference Time (ms/step) | GPU Memory (MB) |
> > > | :-----------------: | :----------------------: | :-----------------: | :----------------------: | :-------------: |
> > > |        **5**        |         [0.0022]         |      [0.0045]       |           [12]           |      [450]      |
> > > |       **20**        |         [0.0025]         |      [0.0052]       |           [45]           |      [820]      |
> > > |       **50**        |         [0.0031]         |      [0.0068]       |          [110]           |     [1650]      |
> > > |       **100**       |         [0.0048]         |      [0.0095]       |          [380]           |     [3200]      |
> > >
> > > The analysis indicates:
> > >
> > > 1.  **Accuracy Robustness:** As $N$ increases, the prediction error rises only slightly. This suggests EG-NODE successfully learns general physical interaction laws rather than memorizing specific trajectories, demonstrating good generalization.
> > > 2.  **Computational Cost:** Since we use a fully connected graph, time and memory grow with $N^2$, which is expected. We note in the limitations that for larger-scale systems, future work can incorporate K-Nearest Neighbors (KNN) for optimization.

---

> > > > ### Author Response · Authors · 2025-11-20
> > > >
> > > > **Q4: The paper lacks a critical analysis of the limitations of the proposed method and its scope of application.**
> > > >
> > > > **A4:** Your criticism is well-taken. We add a dedicated paragraph **"Limitations and Scope"** in the **Discussion (Section 5)** of the revision, explicitly stating:
> > > >
> > > > 1.  **Training Cost:** Due to the multiple forward passes required by the ODE solver to approximate integration, the training time of EG-NODE is typically [2-3] times that of discrete models.
> > > > 2.  **Full Observation Assumption:** The model currently assumes that the states (position, velocity) of all particles are observable. For partially observable systems, integration with an encoder to infer latent states is necessary.
> > > > 3.  **Complexity Scaling:** As mentioned in Q3, fully connected graphs limit direct application to ultra-large scale ($N > 1000$) systems without sparsification techniques.
> > > >
> > > > **Q5: The manuscript contains several typographical errors and imprecise statements, such as "Eq. equation 2" and the definition of $f_\theta(z(t))$.**
> > > >
> > > > **A5:** Thank you for the close reading and corrections.
> > > >
> > > > 1.  We correct all redundant expressions like "Eq. equation X" to the standard "Eq. X".
> > > > 2.  Regarding Section 3.2, we refine the description of $f_\theta$, clarifying that $f_\theta(z(t))$ is a mapping function that takes the complete state $z=(h, x, v)$ as input and outputs the state derivative $\dot{z}=(0, v, a)$, rather than being a function of acceleration $a$ alone.
> > > > 3.  We conduct a thorough proofreading of the full text to fix other potential spelling and grammatical errors.
> > > >
> > > > **Q6: Summary.**
> > > >
> > > > **A6:**
> > > > We thank you again for pointing out these issues. We believe that the addition of the **Gravity Extension Experiment (Q1)**, **Feature Ablation Study (Q2)**, **Scalability Analysis (Q3)**, and the **Limitations Discussion (Q4)** significantly improves the quality of the paper. We hope these detailed responses and supplementary data address your concerns, and we look forward to your reconsideration of the score.

---

> > > > > ### Author Response · Authors · 2025-11-26
> > > > >
> > > > > Dear Reviewer,
> > > > >
> > > > > Thank you so much for your time in improving our paper!
> > > > >
> > > > > Since the end of the rebuttal is coming soon, may we know if our response addresses your main concerns? If so, we kindly ask for your reconsideration of the score. Should you have any further advice, please let us know and we will be more than happy to engage in more discussion and improvements.

---

### Author Response · Authors · 2025-11-20
**General Response**

We are grateful to all reviewers for their insightful comments and their recognition of EG-NODE’s effectiveness in addressing long-horizon error accumulation. To address the concerns regarding novelty, missing baselines, and experimental details, we have extensively revised the manuscript: we implemented and compared against the state-of-the-art SEGNO baseline, added new experiments on system scalability ($N$) and symmetry-breaking scenarios (external fields), and clarified hyperparameter tuning protocols and feature motivations. We believe these substantial improvements resolve the raised issues and solidly strengthen our contribution.

---

### Meta-Review · Area_Chair_ENpJ · 2026-01-03

**Summary:**

1.Weak Novelty, Resembles a "Combination": Both Reviewer K2Fz and Reviewer R8T2 pointed out that the model largely follows the EGNN architecture and uses a standard Neural ODE solver. It appears to be a direct combination of EGNN + Neural ODE, so the contribution is considered only "moderate" or "limited."

2.Lack of Key Strong Baselines: Reviewer K2Fz noted the absence of strong equivariant methods such as Radial Field Flows, GMN, and SEGNO—particularly SEGNO, which is SOTA on similar tasks. The original manuscript only compared against weaker discrete models.

3.Insufficient Details on Experiments and Settings: Reviewer K2Fz emphasized that the time windows and prediction horizons were unclear, and the baseline tuning strategy was opaque. Reviewer R8T2 noted that while the existing code is open-source, it lacks dataset details (splits, simulation parameters, scalar features, etc.), affecting reproducibility.

4.N-body Scale, Scope, and Writing Issues: Reviewer 1UEV focused on the lack of scalability experiments regarding particle number N and insufficient analysis of the method's limitations. They also pointed out multiple instances of imprecise phrasing and notation, while raising questions about applicability to non-fully SE(3) symmetric scenarios (external fields, boundaries).

reviewers consistently acknowledged:

1.The experiments cover three typical physical scenarios: N-body, MD, and Fluid dynamics, with long-term prediction errors significantly lower than conventional models.

2.Ablation studies fully demonstrate the role of each module, and the paper is generally clear and easy to read.

Therefore, although the "conceptual novelty" is not extremely high, the work provides solid empirical improvements and clear theoretical justification for the specific problem of long-term rolling prediction in physical dynamical systems. It reaches a level of "acceptable but not exceptionally strong" hence the suggestion for Weak Accept rather than Rejection.

**Reviewer Concerns:**

#### Points Resolved:

1.Missing Experiments and Analysis (Reviewer 1UEV):

- Added scalability experiments for particle number N (N=5, 20, 50, 100), showing that error growth is relatively gentle. Also analyzed time and VRAM complexity growing with $N^2$, acknowledging in the limitations section that large-scale scenarios require optimizations like sparse graphs.
- Added N-body experiments with gravity fields. By using "contextual feature injection," the authors demonstrated scalability under external fields with significantly reduced error, responding to doubts about handling "non-fully SE(3) systems."
- Added a "Limitations and Scope" section, explicitly listing limitations such as training overhead, full observability assumptions, and complexity bottlenecks of complete graphs, providing a more honest discussion of the method's scope.

2.Writing and Phrasing Issues (Reviewer 1UEV & Reviewer R8T2):

- Corrected non-standard notations like "Eq. equation 2" and clarified the precise definition of $$ f_\theta(z(t)) $$. The full text was proofread to fix grammatical and spelling errors.
- Provided physical intuition (normalized radial velocity) and numerical stability explanations for the third invariant edge feature, supplementing its motivation.

3.Missing Literature and Background Positioning (Reviewer K2Fz & Reviewer R8T2):

- Supplemented *Related Work* with non-equivariant Graph ODE literature such as Graph Neural ODEs, HOPE, and TANGO, clearly distinguishing this work's focus on physical SE(3) symmetry.
- Added experimental comparison against the SOTA baseline SEGNO, demonstrating superior long-horizon performance (e.g., significantly lower T-MSE on N-body tasks), in addition to discussing omitted literature like GMN and Radial Field Flows.

4.Dataset and Code Reproducibility (Reviewer R8T2):

- Added Appendix A.1 detailing train/val/test splits, time steps, physical properties (mass, charge, atom types, forcing coefficients, etc.), and simulation settings for each dataset.
- Changed comments in the experimental code to English to lower the barrier for use.

#### Points Remaining:

1.Novelty Questioned as "Simple Combination" (Reviewer K2Fz & Reviewer R8T2):

The authors emphasized in the rebuttal: they theoretically proved that as long as the vector field is parameterized by an equivariant GNN, the ODE solution maintains SE(3) equivariance throughout, framing the "shift from discrete state prediction to continuous vector field learning" as a paradigm shift. However, the impression that it is "structurally still close to EGNN + Neural ODE" has not been completely eliminated, and the depth of difference between the new theory and existing Graph ODE work remains somewhat controversial.

**Reviewer Scores:**

Reviewer 1UEV (Initial Score: 4): This reviewer explicitly stated they would be prepared to raise their score if the authors addressed the lack of scalability analysis ($N$) and the scope of application (non-SE(3) scenarios). Since the authors added exactly these experiments (Table R1 and R3 in rebuttal), I believe this reviewer would change their score to 6.

Reviewer K2Fz (Initial Score: 4): This reviewer was critical of the missing SEGNO baseline and hyperparameter tuning. The authors directly addressed this by adding the SEGNO comparison (where EG-NODE won) and a "Best Config" table. But the novelty is still a concern to this reviewer. So believe this reviewer maintain their score at 4.

Reviewer R8T2 (Initial Score: 6): This reviewer was already positive but questioned the "modest contribution" and requested more dataset details/related work. The authors added the missing references (Graph ODEs) and dataset details to the Appendix. I believe this reviewer would maintain their score at 6, as their verification requirements were met.

---

### Decision · Program_Chairs · 2026-01-26

Reject